# Comparison of COVID-19 and Non-COVID-19 Pneumonia in Down Syndrome

**DOI:** 10.3390/jcm10163748

**Published:** 2021-08-23

**Authors:** Diego Real de Asua, Miguel A. Mayer, María del Carmen Ortega, Jose M. Borrel, Teresa de Jesús Bermejo, Domingo González-Lamuño, Coral Manso, Fernando Moldenhauer, María Carmona-Iragui, Anke Hüls, Stephanie L. Sherman, Andre Strydom, Rafael de la Torre, Mara Dierssen

**Affiliations:** 1Adult Down Syndrome Outpatient Unit, Department of Internal Medicine, Hospital Universitario de La Princesa, 28006 Madrid, Spain; diego.realdeasua@gmail.com (D.R.d.A.); fernando.moldenhauer@salud.madrid.org (F.M.); 2Instituto de Investigación Sanitaria La Princesa, Hospital Universitario de La Princesa, 28006 Madrid, Spain; 3Research Programme on Biomedical Informatics, Hospital del Mar Medical Research Institute, 08003 Barcelona, Spain; miguelangel.mayer@upf.edu; 4Department of Experimental and Health Sciences, Universitat Pompeu Fabra, 08003 Barcelona, Spain; rtorre@imim.es; 5Department of Psychiatry, Research Institute i+12, Hospital Universitario 12 de Octubre, 28041 Madrid, Spain; maica.otrb@gmail.com; 6Down España Huesca, 22005 Huesca, Spain; pepeborrel@telefonica.net; 7Unidad de Neuropediatría, Instituto Hispalense de Pediatría, 41013 Sevilla, Spain; teresa_bermejo@hotmail.com; 8Department of Pediatrics, University Hospital Marqués de Valdecilla-Universidad de Cantabria, 39008 Santander, Spain; domingo.gonzalez-lamuno@unican.es; 9Down España, 28043 Madrid, Spain; salud@sindromedown.net; 10Sant Pau Memory Unit, Department of Neurology, Hospital de la Santa Creu i Sant Pau, Biomedical Research Institute Sant Pau, Universitat Autònoma de Barcelona, 08025 Barcelona, Spain; mcarmonai@santpau.cat; 11Barcelona Down Medical Center, Fundació Catalana de Síndrome de Down, 08029 Barcelona, Spain; 12Department of Epidemiology and Gangarosa Department of Environmental Health, Rollins School of Public Health, Emory University, Atlanta, GA 30322, USA; anke.huels@emory.edu; 13Department of Human Genetics, School of Medicine, Emory University, Atlanta, GA 30322, USA; ssherma@emory.edu; 14Department of Forensic and Neurodevelopmental Sciences, Institute of Psychiatry, Psychology, and Neuroscience, King’s College London, London SE5 8AF, UK; andre.strydom@kcl.ac.uk; 15The London Down Syndrome (LonDownS) Consortium, London, United Kingdom and South London and the Maudsley NHS Foundation Trust, London SE5 8AF, UK; 16Neurosciences Research Programme, Hospital del Mar Medical Research Institute, 08003 Barcelona, Spain; 17Biomedical Research Networking Center for Physiopathology of Obesity and Nutrition (CIBEROBN), 08003 Barcelona, Spain; 18Center for Genomic Regulation, The Barcelona Institute for Science and Technology, 08003 Barcelona, Spain; 19Biomedical Research Networking Center for Rare Diseases (CIBERER), 08003 Barcelona, Spain

**Keywords:** down syndrome, COVID-19, SARS-CoV-2, pneumonia

## Abstract

Whether the increased risk for coronavirus disease 2019 (COVID-19) hospitalization and death observed in Down syndrome (DS) are disease specific or also occur in individuals with DS and non-COVID-19 pneumonias is unknown. This retrospective cohort study compared COVID-19 cases in persons with DS hospitalized in Spain reported to the Trisomy 21 Research Society COVID-19 survey (*n* = 86) with admissions for non-COVID-19 pneumonias from a retrospective clinical database of the Spanish Ministry of Health (*n* = 2832 patients). In-hospital mortality rates were significantly higher for COVID-19 patients (26.7% vs. 9.4%), especially among individuals over 40 and patients with obesity, dementia, and/or epilepsy. The mean length of stay of deceased patients with COVID-19 was significantly shorter than in those with non-COVID-19 pneumonias. The rate of admission to an ICU in patients with DS and COVID-19 (4.3%) was significantly lower than that reported for the general population with COVID-19. Our findings confirm that acute SARS-CoV-2 infection leads to higher mortality than non-COVID-19 pneumonias in individuals with DS, especially among adults over 40 and those with specific comorbidities. However, differences in access to respiratory support might also account for some of the heightened mortality of individuals with DS with COVID-19.

## 1. Introduction

The SARS-CoV-2 coronavirus outbreak causing the coronavirus disease 2019 (COVID-19) affects especially elderly individuals and those with chronic medical conditions. Individuals with Down syndrome (DS) appear to be particularly vulnerable to the disease, with a four-fold increased risk for COVID-19-related hospitalization and a reported increased risk for COVID-19-related mortality from 3- to 10-fold [1,2,3]. People with DS have specific socio-demographic risk factors for COVID-19 and frequently suffer from comorbidities, such as obesity, diabetes mellitus, congenital heart disease, and respiratory diseases, associated with a poorer COVID-19 prognosis in the general population [4]. DS genetic factors, such as up-regulation of the transmembrane protease, serine 2 (*TMPRSS2*) gene, and inflammatory genes [5,6], could also contribute to heightened risk for SARS-CoV-2 infection and worse clinical outcomes. Indeed, mortality rates are increased in adults with DS compared the general population [7] even after adjusting for known risk factors for COVID-19 mortality [2,4,8]. However, because over-expression of TMPRSS2 explains the greater impact of viral diseases (influenza, SARS-CoV, metapneumovirus, MERS) that use ACE-2-receptor/TMPRSS2 for cell binding/cell entry on patients with DS, it is unknown to what extent the increased clinical vulnerability recently observed in DS is specific for COVID-19 [9]. 

Individuals with DS develop more severe complications after contracting respiratory viruses, such as influenza or respiratory syncytial virus [10,11,12], with increased risk of hospital admission, more frequent superimposed bacterial pneumonias, and greater risk of intubation and mortality. Establishing whether increase in mortality is COVID-19-specific could play a crucial role in guiding institutional policies and significantly influence care recommendations for this population. 

Here, we compared a case series of individuals with DS infected with COVID-19 in Spain with a retrospective clinical database of the Spanish Ministry of Health to characterise their clinical presentation and compare their outcomes to cases of non-COVID-19 pneumonias. Given that the deficiencies in immune response have been claimed to be the major contributing factor in increased COVID-19 susceptibility in DS, we included in our control cohort both viral and bacterial pneumonias, which are known to have a higher mortality risk in adults with DS. 

## 2. Materials and Methods

### 2.1. Study Design and Participants

This is a retrospective study including the Spanish cases of a larger international study on COVID-19 in individuals with DS led by the Trisomy 21 Research Society (T21RS, https://www.t21rs.org/ data collected from 15 March 2020 to 31 July 2020 [8]). The study is based on online surveys administered to caregivers/family members of COVID-19-affected individuals with DS (family surveys) and to clinicians (clinician surveys), capturing similar information in 34 questions, including socio-demographic data, living situation during the pandemic, pre-existing medical and psychiatric conditions, SARS-CoV-2 diagnostic tests, COVID-19 signs and symptoms, hospital and ICU admission, and clinical outcome. The clinician survey included additional information about complications, medications, and treatments used during COVID-19 illness. The surveys were developed in March 2020, implemented through REDCap [13,14], and hosted at Emory University (Atlanta, GA, USA). In our study, we included individuals with DS hospitalized with confirmed diagnoses of COVID-19 (either through a positive RT-PCR determination in a nasopharyngeal swab or a positive IgM and/or IgG serological test) as well as those without confirmatory testing but with symptoms suggestive of COVID-19. All Spanish cases reported from 15 March to 31 July 2020 (first lockdown period) were included. COVID-19 patients under 16 years of age were exceptional in the T21RS clinical survey (only 3/89). Thus, we limited the study to patients older than 15. After excluding patients with missing relevant information, such as age, gender, or clinical information, our sample included data from 86 hospitalized Spanish COVID-19 patients with DS (T21RS cohort). 

For our retrospective control cohort (CMBD cohort), we used data derived from a large clinical database of the Spanish Ministry of Health, the Minimum Basic Dataset (*Conjunto Mínimo Básico de Datos*, CMBD; https://www.mscbs.gob.es/estadEstudios/estadisticas/estadisticas/estMinisterio/SolicitudCMBDdocs/Formulario_Peticion_Datos_CMBD.pdf; accessed on 31 July 2020). CMBD collects information about relevant clinical aspects, including length of stay, comorbidities, procedures, in-hospital mortality, and outcomes of all admissions to hospitals and primary care centres in Spain. Each admission is classified into a diagnosis-related group, allowing the comparison of patients with clinically similar events. 

### 2.2. Data Extraction

We extracted all consecutive hospital admissions found in CMBD from 2005 to 2014 that included the ICD-9-CM code 758.0 (Down syndrome) as main or secondary diagnosis and one of the following bacterial or viral pneumonia ICD-9-CM codes: 480.1 (pneumonia due to Respiratory Syncytial Virus), 481(Pneumococcal pneumonia), 483.0 (pneumonia due to Mycoplasma pneumoniae), 485 (bronchopneumonia, unspecified organism), 486 (pneumonia, unspecified organism), and 488.1 (influenza due to identified 2009 H1N1 influenza virus). Our CMBD cohort consisted of 2832 admissions of individuals with DS older than 15 years of age admitted between 2005 and 2014 to Spanish hospitals for any of the aforementioned causes of pneumonia. We also evaluated three relevant clinical comorbidities: obesity, dementia, and epilepsy/seizures, due to their association with worse outcomes for respiratory diseases, including COVID-19, in DS patients [15]. For the identification of these comorbidities within the CMBD database, the following diagnostic codes were used: obesity: 278.0, 278.00, 278.01, 278.02 and 278.03, V85.4, V85.41, and V85.42; dementia: 331.0, 331.1, 331.9, 290.0, 290.10, 290.11, 290.12, 290.13, 290.2, 290.20, 290.21, 290.3 and 290.4, 294.1, 294.10, 294.11, 294.20, and 294.21; epilepsy: 345.0, 345.00, 345.01, 345.1, 345.10, 345.2, 345.3, 345.4, 345.5, 345.6, 345.7, 345.8 345.9, and 345.90.

### 2.3. Statistical Analysis

We used descriptive statistics to show the demographic information, outcomes, COVID-19 symptoms, and comorbidities of the participants included in our analyses. Categorical data were described as frequencies (percentages), and quantitative variables are provided as the mean (SD) or median (IQR). The χ^2^ test or the Fisher’s exact test were used for comparisons involving qualitative variables, and Student’s *t*-test or Mann–Whitney U test (when applicable) were used for quantitative variables. To account for possible age-related differences between the two cohorts, we also stratified the results for age. Furthermore, we estimated an age-standardized mortality ratio and estimated the relative risk of death between patients with COVID-19 and patients with viral and bacterial pneumonias, with the associated 95% CI.

Associations between potential risk factors and all-cause mortality after a diagnosis of COVID-19 were analyzed using adjusted logistic regression models. Associations with age and sex were adjusted for data source (family vs. clinician surveys), and associations with living situation, level of intellectual disability, and clinical and mental comorbidities were adjusted for age, sex, and data source. The statistical significance threshold was set at *p* < 0.05 and two-sided. We performed all data analyses using R (version 4.0.0, R Core Team (2021), Vienna, Austria) and Jamovi (version 1.2.24.0, Jamovi project (2021), Sydney, Australia). 

## 3. Results

### 3.1. Socio-Demographic and Clinical Characteristics of the Spanish DS COVID-19 Patients

A total of 150 Spanish patients with DS and COVID-19 were reported to the T21RS survey [15], 42 (28%) of which were reported by families and 108 (72%) by clinicians. The main socio-demographic characteristics of the cohort are summarised in Appendix A. There were no statistically significant differences in the distribution by sex, type of trisomy, or level of intellectual disability between family and clinician surveys. Cases reported by families were younger than those reported by clinicians (*p* < 0.001; Appendix A). Regarding comorbidities, only behavioural problems (29 of 108 vs. 1 of 42; *p* < 0.001) and Alzheimer’s disease (33 of 100 vs. 1 of 34; *p* < 0.001) were significantly more reported by clinicians. The prevalence of relevant comorbidities reported in the family and clinician surveys are shown in Appendix A. Regarding COVID-19 symptoms, fever, cough, and shortness of breath were the most frequently reported symptoms in both groups (Appendix A). 

### 3.2. Comparison of COVID-19 and Other Pneumonia-Related Hospitalizations

For the comparison with the non-COVID-19 pneumonias, from the 150 patients with DS, we included the 86 older than 15 years infected by SARS-CoV-2 who were hospitalized due to COVID-19 (86/150; 59.3%). The clinical presentation of the hospitalized subgroup did not differ from those of the T21RS cohort ([15], Appendix A), and their pharmacological treatment during hospitalization is summarised in Appendix A. The CMBD cohort consisted of 2832 patients with DS hospitalized for non-COVID-19 pneumonias. The distribution by age group and the general characteristics of hospitalized DS patients with COVID-19 and non-COVID-19 pneumonias is presented in Table 1. 

DS patients admitted for COVID-19 were slightly older than those admitted for non-COVID-19 pneumonias (T21RS cohort vs. CMBD cohort: 46.4 years (SD 11.8) vs. 42.2 years (SD 13.9), *t* = 3.224, *p* < 0.01), although this difference was not clinically significant. No differences were detected in sex distribution between COVID-19 and non-COVID-19 pneumonias. In-hospital mean length of stay of DS patients with COVID-19 was similar to that of non-COVID-19 patients (10.3 days (SD 7.9) vs. 9.9 (SD 9.4), *t =* 0.264, *p* = 0.79). However, there was a noticeable difference in length of stay by discharge status. DS patients with COVID-19 who were discharged alive stayed significantly longer than those who died during their admission (12.1 days (SD 8.2) vs. 5.1 days (SD 3.4), *t =* 5.39; *p* < 0.001), whereas the opposite was detected in patients with DS admitted for non-COVID-19 pneumonias, suggesting a more severe course of disease in COVID-19 patients. Even so, the mean length of stay of COVID-19 DS patients who died while hospitalized was significantly shorter than that of non-COVID-19 cases (*t* = −5.95, *p* < 0.001). The distribution of patients according to age, length of admission, and outcome in both groups is depicted in Figure 1 (T21RS cohort) and Appendix A (CMBD cohort). 

With regard to the presence of relevant comorbidities, the frequency of a prior medical history of epilepsy among hospitalized DS patients with COVID-19 was similar to that observed in non-COVID-19 pneumonias. However, a significantly higher proportion of COVID-19 patients had dementia (25.5% vs. 5.4% *p* < 0.001) or obesity (25.5% vs. 8.7% *p* < 0.001), which have been related to risk of COVID-19 severity (Table 2).

These differences could not be attributed to differences in age or outcome, as both groups have similar profiles (Appendix A).

### 3.3. Mortality Rates 

The overall in-hospital mortality rate for DS patients with COVID-19 was 26.7% (23 of 86), ranging from 37% (23 of 62) among those over 40 years to 0.0% (0 of 24) among those under 40 (Table 3 and Appendix A), with the odds ratio (OR) for mortality of DS adults over 40 = 18.4 (CI 95% 3.7–334.8). 

In DS patients with non-COVID-19 pneumonias, the mortality rate was 9.7% (273 of 2811), ranging from 13.7% (214 of 1567) in those over 40 years to 4.7% (59 of 1244) in patients under 40 (Table 3 and Appendix A). The mean age of deceased patients with COVID-19 was 55.5 ± 5 years, while in the CMBD cohort, it was 49.1 ± 11.7 years (*p* < 0.001). A regression analysis to determine whether differences in mortality rates were age-related showed that increased mortality of COVID-19 patients (OR 2.8, CI 95% 1.7–4.7) was still detected even after adjusting for age and sex.

Logistic regressions to evaluate the association between mortality reported comorbidities and socio-demographic risk factors were only performed in the DS-COVID-19 cohort, as we do not have this information in the non-COVID-19 cases (Figure 2). 

Obstructive sleep apnea (OR 1.5, 95%—IC 0.4–6.1), global psychiatric condition (OR 0.7—CI 0.2–2.3), behavioral problems (OR CI 1.2, 95%—IC 0.3–4.5), seizures (OR 8.3—CI 1.2–45), Alzheimer’s disease (OR 3.4—CI 0.8–12.7), and age (OR 1.2—CI 1.1–1.3) presented a positive association with mortality, although only age and seizures reached statistical significance. Nevertheless, when seizures were adjusted for the presence of Alzheimer’s disease, their association with mortality was no longer statistically significant. 

COVID-19 patients with dementia presented a mortality rate of 63.6% (14 of 22), those with obesity of 22.7% (5 of 22), and those with epilepsy 77.8% (7 of 9). These rates were significantly higher than those observed in non-COVID-19 pneumonias, where only 19.7% (30 of 152) patients with dementia, 6.6% (19 of 246) patients with obesity, and 15.2% (57 of 389) patients with epilepsy died (Fischer’s exact test *p* < 0.001, *p* = 0.03, and *p* < 0.001, respectively).

## 4. Discussion

A growing body of evidence points to a more severe impact of COVID-19 on individuals with Down syndrome (DS) compared to the general population. We here confirm that the mortality rate for COVID-19 infection is significantly higher than that for non-COVID-19 pneumoniae in individuals with DS. Our results suggest that this increased mortality rate could be attributed to an increased prevalence of COVID-19-related comorbidities, but they also reveal differences in access to treatment options, such as respiratory support, that might account for the heightened mortality in COVID-19-DS patients.

Compared to euploid controls, hospitalized COVID-19 patients with DS are younger and show a higher prevalence of obesity, diabetes, and dementia even accounting for differences in dementia detection and diagnosis in primary care [16,17]. Instead, when compared with hospitalized adults with DS for non-COVID-19 pneumonias, our analysis revealed that those hospitalized for COVID-19 are older and present a higher prevalence of obesity, dementia, and/or seizures.

Regarding mortality due to COVID-19, the estimates vary across studies. Clift et al. [2] and Malle et al. [18] reported a 10-fold increased risk for COVID-19-related death in DS, while Hüls et al. [15] reported a 2.9 to 3.5-fold increase. Comparing COVID-19 with other respiratory pathogens in individuals with DS, we detected 2.8-fold higher mortality due to COVID-19. Age showed a strong positive association with mortality so that COVID-19-hospitalized patients over 40 had around 18 times more chances of dying than those under 40, with a mortality rate of 37%. This is close to three times higher than that observed for other pneumonias in this same age group even though our control cohort also purposely included severe bacterial pneumonias, which have a higher mortality risk in adults with DS [19,20]. Recent analyses in the general population also demonstrated higher mortality rates in SARS-CoV-2 than other respiratory pathogens [21,22,23,24]. However, it has to be borne in mind that the ageing process in DS subjects is premature, leading to high levels of mortality and multi-morbidity in this population [25].

In adults with DS, differences in the prevalence of COVID-19-related comorbidities, altered immune regulation, and inflammation could explain the increased mortality [5,26]. We confirmed a disproportionate mortality burden of obesity, dementia, and epilepsy, all of which are well-known mortality risk factors for COVID-19 in the general population. In our cohort, cases with COVID-19 and dementia presented a mortality rate of 63.6% compared to 19.7% for non-COVID-19 pneumonias, those with epilepsy 77.8% versus 15.2%, and those with obesity of 22.7% versus 6.6%.

However, these comorbidities do not fully explain the increased risk of death of COVID-19 seen in DS. One aspect contributing to the excess mortality risk might be differences in therapy, but our cohort received pharmacological treatments, such as glucocorticoids (Appendix A), in a similar rate as euploid COVID-19 Spanish patients [27,28]. An interesting finding of our study, though, is that the in-hospital length of stay was significantly shorter in adults with DS who died due to COVID-19 than those who survived or those with non-COVID-19 pneumonias. One could speculate that, in the first wave of the pandemic, individuals with DS may have been admitted to hospitals later due to delays in the diagnosis, leading to a worse clinical prognosis. However, this trend has not been observed in the general population admitted for SARS-CoV-2 pneumonia [24,29]. Another explanation could lie in the patient surge suffered by ICUs during the first lockdown period, which might have limited the ICU access equally to DS and other patients. Indeed, the rate of hospitalized patients with DS admitted to the ICU (4.3%) was significantly lower than those reported for the general population [24], suggesting unequal access to ventilatory support. The correlation we found between age and age-related comorbidities (Alzheimer’s disease and seizures) with a shorter length of stay and a higher rate of fatal outcomes strengthens the possibility that differences in access to treatment options, such as respiratory support, might have accounted for the heightened mortality in COVID-19 in DS patients. 

Our study has several limitations. First, the T21RS survey was launched when available testing for the SARS-CoV-2 was minimal and the major signs and symptoms related to COVID-19 were still being catalogued. Second, we compared a survey to an administrative database, which does not contain the same level of information. Additionally, the use of questionnaires has recollection and report bias. In addition, many asymptomatic cases may not be reported, and thus the true denominator of all admissions of adults with DS is not known, and mortality rates might be overestimated. Finally, we recognize that our control group includes many different types of pneumonia, with differing morbidity and mortality. Indeed, the comparison with other viral pneumonias is pathophysiologically appropriate given that patients with DS are more prone to viral diseases that use the ACE-2-receptor/TMPRSS2 for cell binding/cell entry. However, the comparison with bacterial pneumonias is also relevant given that these have higher morbidity and mortality rates than their viral counterparts both in the general population and in adults with DS. Thus, a control group that includes both types of pneumonias helps to strengthen our thesis that SARS-CoV-2 infection carries an excess mortality risk greater than other respiratory pathogens.

## 5. Conclusions

Our study confirms that in DS individuals, the mortality rate for COVID-19 infection is significantly higher than that for other non-COVID-19 pneumoniae. Although this greater impact of SARS-CoV-2 compared to other pathogens could be attributed to a particular susceptibility to infection due to impaired host responses in DS, our results hint that differences in access to treatment options, such as ventilatory support, might account for the heightened mortality in COVID-19. Our findings highlight the importance of reinforcing appropriate preventive measures against COVID-19 for individuals with DS as well as the urgent need for a specific vaccine and treatment for DS patients.

## Figures and Tables

**Figure 1 jcm-10-03748-f001:**
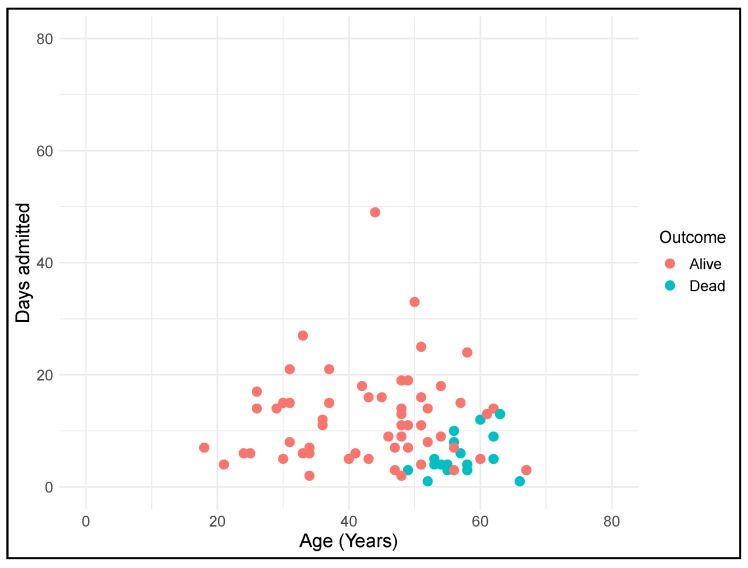
Distribution of hospitalization length of stay for individuals with DS with COVID-19 (T21RS cohort) according to age and outcome. Data extracted only from clinicians’ surveys.

**Figure 2 jcm-10-03748-f002:**
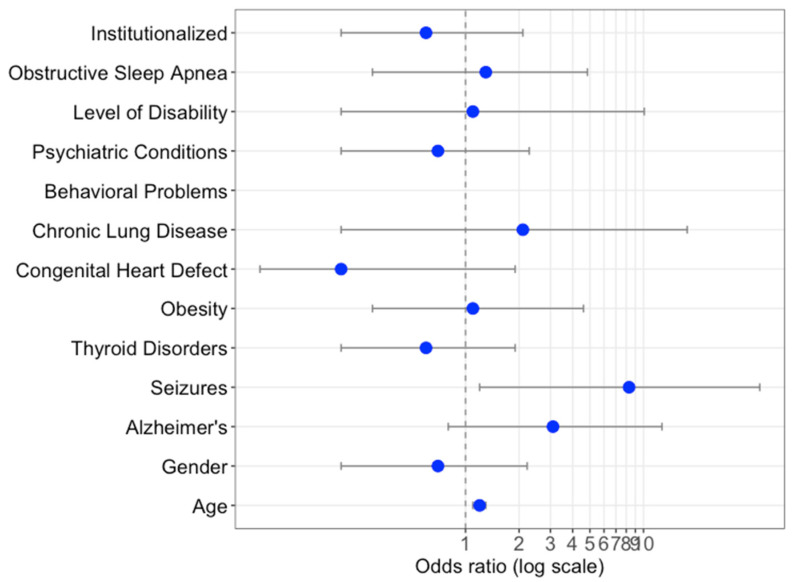
Risk factors and their association with mortality in patients with DS and COVID-19. Age and gender were adjusted by survey, and all of the risk factors were adjusted by age, gender, and source of data (family or clinician survey).

**Table 1 jcm-10-03748-t001:** General characteristics of Spanish DS patients older than 15 years admitted to the hospital due to COVID-19 or non-COVID-19 pneumonia.

	COVID-19 Pneumonia	Non-COVID-19 Pneumonia	*p*
	*n* = 86	*n* = 2832	
Age (years)	46.4 (11.8)	42.2 (13.9)	<0.01 *
(mean (SD))			
16–40 years			0.36 **
Women	24 (27.9%)	1251 (44.2%)	
Men	11 (45.8%)	434 (34.7%)	
(*n* (%))	13 (56.2%)	817 (65.3%)	
>40 years			0.21 **
Women	61 (72.1%)	1581 (55.8%)	
Men	31 (50.8%)	662 (41.9%)	
(*n* (%))	30 (49.2%)	919 (58.1%)	
Length of stay	10.3 (7.9)	9.9 (9.4)	0.79 **
(mean (SD))			
Length of stay by discharge status (Days) (mean (SD))			
Alive	12.1 (8.2)	9.8 (8.6)	<0.001 *
Dead	5.1 (3.3)	11.9 (14.5)	0.034 *

Data are presented as mean (SD) or *n* (%). * Student’s *t*-test, ** chi-square test.

**Table 2 jcm-10-03748-t002:** Frequency of obesity, dementia, and epilepsy comorbidities in individuals with Down syndrome (DS) and COVID-19 compared to individuals non-COVID-19 pneumoniae.

	DS and COVID-19 *n* = 86 *n* (%)	Non-COVID Pneumoniae *n* = 2832 *n* (%)	χ^2^ (*p*)
Obesity	22 (25.5)	246 (8.7)	<0.001
Dementia	22 (25.5)	152 (5.4)	<0.001
Epilepsy	9 (10.4)	389 (13.7)	0.7

DS, Down syndrome; results are presented as *n* (%).

**Table 3 jcm-10-03748-t003:** Mortality rates in DS patients with COVID-19 and with other non-COVID-19 pneumoniae.

	COVID-19 Pneumonia *n* = 86	Non-COVID Pneumoniae *n* = 2811	*p*
Overall in-hospital mortality rate (*n* (%))	23 (26.7)	273 (9.7)	<0.001 *
Age of deceased patients (mean (SD))	55.5 (5)	49.1 (11.7)	<0.001 **
Proportion of deceased patients over 40 years (*n* (%))	23 (37)	214 (13.7)	<0.001 *

Data are presented as mean (SD) or *n* (%). * chi-square test, ** Student’s *t*-test.

## Data Availability

This study involves the use of patient medical data from the Spanish Centralized Hospital Discharge Database (CMBD). CMBD data are hosted by the Ministry of Health, Consumption and Social Welfare (MSCBS). Researchers working in public and private institutions can request the databases by filling, signing and sending a questionnaire available at the MSCBS website. In this questionnaire, a signed Confidentiality Commitment is required. All data are anonymized and de-identified by the MSCBS before it is provided to applicants. According to this Confidentiality Commitment signed with the MSCBS, researchers cannot provide the data to other researchers that must request the data directly to the MSCBS in the following link: https://www.mscbs.gob.es/estadEstudios/estadisticas/estadisticas/estMinisterio/SolicitudCMBDdocs/Formulario_Peticion_Datos_CMBD.pdf. The T21RS dataset is managed and maintained by the Department of Epidemiology and Gangarosa Department of Environmental Health, Rollins School of Public Health, Emory University, Atlanta. There is no direct access to the T21RS database, but specific requests can be made to the society.

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
