# Peer review of "Comparison of COVID-19 and Non-COVID-19 Pneumonia in Down Syndrome"

_jcm, 2021, doi:10.3390/jcm10163748_

Round 1

Reviewer 1 Report

I have read with interest the paper "Comparison of COVID-19 and non-COVID-19 pneumonia in Down syndrome", I suggest the publication after some clarification.
It is not clear why you include in the comparison analysis only 86 pts from initial 150 pts
Why do you choose to compare a viral pneumonia with bacterial pneumonia? even in normal population the outcome is different. I think to highlight the result of the study would be better to perform a sub analysis focused on viral pneumonia vs. covid pneumonia
Regarding the difference in mortality could you speculate that in the first wave physicians where not ready to face the covid emergency and even the treatment was different among centers. It would be interesting in the future to compare the mortality in DS pts in first vs. second waves and so on.
The difference in mortality was observed after 40 yo old I think is better to add a sentence where you explain which means 40 yo for a DS person (for example 55 yo is close to 60 yo that is like 80 yo for non DS people, it is close the end of life). I suggest to read the paper "Vita, S., Di Bari, V., Corpolongo, A., Goletti, D., Espinosa, J., Petracca, S., ... & Valli, M. B. (2021). Down Syndrome patients with COVID-19 pneumonia: A high-risk category for unfavourable outcome. International Journal of Infectious Diseases, 103, 607-610."

Author Response

Response to Reviewer 1

I have read with interest the paper "Comparison of COVID-19 and non-COVID-19 pneumonia in Down syndrome", I suggest the publication after some clarification.

Point 1. It is not clear why you include in the comparison analysis only 86 pts from initial 150 pts.

Response 1. We compared patients that were admitted to hospital. So in order to compare both groups of patients that were admitted to hospital (non-covid pneumonia and covid pneumonia) we excluded from the 150 patients the 64 patients who did not require admission to the hospital. We also excluded patients under 16 years of age as those were exceptional in the T21RS clinical survey (only 3 from these 89) obtaining 86 patients. We have now changed the text to make it clearer. “For the comparison with the non-COVID-19 pneumonias, from the 150 patients with DS we included the 86 older than 15 years infected by SARS-CoV-2 who were hospitalized due to COVID-19 (86/150;59.3%).” See lines 176-179. We also modified the text in lines 106-107 for consistency: “COVID-19 patients under 16 years of age were exceptional in the T21RS clinical survey (only 3/89). Thus, we limited the study to patients older than 15.”

Point 2. Why do you choose to compare a viral pneumonia with bacterial pneumonia? even in normal population the outcome is different. I think to highlight the result of the study would be better to perform a sub analysis focused on viral pneumonia vs. covid pneumonia

Response 2. We thank the reviewer for this comment. Indeed, the comparison with other viral pneumonias is pathophysiologically appropriate given that patients with Down syndrome are more prone to viral diseases (Influenza, SARS-CoV-1, Metapneumovirus, MERS, and SARS-CoV-2) that use ACE-2-receptor/TMPRSS2 for cell binding/cell entry. However, the comparison with bacterial pneumonias is also relevant, given that these have higher morbidity and mortality rates than their viral counterparts both in the general population and in adults with Down syndrome. Thus, a control group that includes both types of pneumonias would only decrease the difference in mortality found between controls and COVID-19 patients. Hence, this compound control group only helps strengthen our thesis. 

We have added this point to the LIMITATIONS section in the DISCUSSION, as follows (lines 339 onward):

In addition, many asymptomatic cases may not be reported, and thus the true denominator of all admissions of adults with DS is not known, and mortality rates might be overestimated. Finally, we recognize that our control group includes many different types of pneumonia, with differing morbidity and mortality. Indeed, the comparison with other viral pneumonias is pathophysiologically appropriate given that patients with Down syndrome are more prone to viral diseases that use the ACE-2-receptor/TMPRSS2 for cell binding/cell entry. However, the comparison with bacterial pneumonias is also relevant, given that these have higher morbidity and mortality rates than their viral counterparts both in the general population and in adults with DS. Thus, a control group that includes both types of pneumonias helps to strengthen our thesis that SARS-CoV-2 infection carries an excess mortality risk greater than other respiratory pathogens.”

Point 3. Regarding the difference in mortality could you speculate that in the first wave physicians where not ready to face the covid emergency and even the treatment was different among centers. It would be interesting in the future to compare the mortality in DS pts in first vs. second waves and so on.

Response 3. Thank you for this important comment. It is completely true that in the first wave physicians, governments, and the healthcare system were facing a new situation and there were many uncertainties from the scientific point of view about the pandemic, factors that could impact the mortality rate. This was included in the discussion (lines 321-324): “One could speculate that, in the first wave of the pandemics individuals with DS may have been admitted to hospitals later due to delays in the diagnosis leading to a worse clinical prognosis. However, this trend has not been observed in the general population admitted for SARS-CoV-2 pneumonia”

In the T21RS taskforce, we have already launched a new survey to collect data from the following waves, and we have also included data from vaccination. We are just starting the process. 

 Point 4. The difference in mortality was observed after 40 yo old I think is better to add a sentence where you explain which means 40 yo for a DS person (for example 55 yo is close to 60 yo that is like 80 yo for non DS people, it is close the end of life). I suggest to read the paper "Vita, S., Di Bari, V., Corpolongo, A., Goletti, D., Espinosa, J., Petracca, S., ... & Valli, M. B. (2021). Down Syndrome patients with COVID-19 pneumonia: A high-risk category for unfavourable outcome. International Journal of Infectious Diseases, 103, 607-610.”

Response 4. We thank the reviewer for pointing this out. We added a sentence to clarify “However it has to be borne in mind that the ageing process in subjects with DS  is premature, leading to high levels of mortality and multi-morbidity in this population.”

Reviewer 2 Report

Nice job and nice presentation.  Some suggestions to improve the study /manuscript.

Introduction needs a little update:

line 65: The first case series reporting on the possible worse outcome of COVID pneumonia was:  De Cauwer H, Spaepen A. Are patients with Down syndrome vulnerable to life-threatening COVID-19? Acta Neurol Belg. 2021 Jun;121(3):685-687. doi: 10.1007/s13760-020-01373-8. Epub 2020 May 22. PMID: 32444942; PMCID: PMC7243430.

The relation between TMPRSS2 and Down (line 69)  (It is assumed that overexpression of chromosome 21 genes, as a result of their presence in an extra copy, causes the Down Syndrome phenotype. In this case, over-expression of TMPRSS2 might explain why Down patients are more prone to viral diseases (Influenza, SARS-CoV-1, Metapneumovirus, MERS, and SARS-CoV-2) that use ACE-2-receptor/TMPRSS2 for cell binding/cell entry. They might get more cells infected because of more protease activity and thus more easily cell entry of the viral pathogens.) was first mentioned in:

De Cauwer H. The SARS-CoV-2 receptor, ACE-2, is expressed on many different cell types: implications for ACE-inhibitor- and angiotensin II receptor blocker-based cardiovascular therapies: comment. Intern Emerg Med. 2020 Nov;15(8):1581-1582. doi: 10.1007/s11739-020-02406-z. Epub 2020 Jun 20. PMID: 32564289; PMCID: PMC7305057.

According to RSV (line 76) there is a recent study of a Dutch group: RSV Gold Study: this study can also be referred to in methods when you discuss the co-morbidities. : Löwensteyn YN, Phijffer EWEM, Simons JVL, Scheltema NM, Mazur NI, Nair H, Bont LJ; RSV GOLD Study Group. Respiratory Syncytial Virus-related Death in Children With Down Syndrome: The RSV GOLD Study. Pediatr Infect Dis J. 2020 Aug;39(8):665-670. doi: 10.1097/INF.0000000000002666. PMID: 32332221; PMCID: PMC7360096.

Materials and methods:

Data extraction: line 120: 'CMBD admissions': 'hospital admissions found in CMBD' sounds better.

line 127: here 16 years is mentioned in line 175 you write 15 years???

line ??? (not mentioned) and line 287: age 40 was already mentioned in other studies: maybe also refer to in materials and methods.

line 129: (see also discussion on TMPRSS2 and RSV above): why not looking for diabetes, osas, heart disease, respiratory disease. diabetes discussed in line 276 but not in methods.

Results:

line 167: more reporting of alzheimers disease due to line 165 older age ?

line 170 a recent study in near future to be published in BMJ Open of a UK based group and already mentioned in:  

https://assets.publishing.service.gov.uk/government/uploads/system/uploads/attachment_data/file/908434/Disparities_in_the_risk_and_outcomes_of_COVID_August_2020_update.pdf

it was shown that people with mental disabilities reported less complaints like anosmia probably leading to late diagnosis of Covid. In fact,  onset of disease or reason for hospitalization was coma,seizures,....(see also line 308-309): please add to discussion.

table 1: please add % in the table, it is very confusing what is SD and what is % . so also change the legend accordingly.

line 199: similar results reported in the UK based study (NHS):  as mentioned later surge capacity, ventilator allocation and end of life decisions might play a significant role.

In the UK  study people with mental disabilities significantly had less ICU admissions and less ventilator access  versus controls.   

Some other last questions/remarks:

  1. do  you have data on how many patients with DS lived with their families or lived in institutions : sick family members  or an outbreak in the institution  might explain longer hospital stay.
  2.  it would be interesting in looking at the same data but with a group of viral pneumonias related  to TMPRSS2 versus other pathogens.

Author Response

Responses to Reviewer 2

Nice job and nice presentation.  Some suggestions to improve the study /manuscript.

We thank the reviewer for the suggestions.

Point 1. Introduction needs a little update:
line 65: The first case series reporting on the possible worse outcome of COVID pneumonia was:  De Cauwer H, Spaepen A. Are patients with Down syndrome vulnerable to life-threatening COVID-19? Acta Neurol Belg. 2021 Jun;121(3):685-687. doi: 10.1007/s13760-020-01373-8. Epub 2020 May 22. PMID: 32444942; PMCID: PMC7243430. The relation between TMPRSS2 and Down (line 69)  (It is assumed that overexpression of chromosome 21 genes, as a result of their presence in an extra copy, causes the Down Syndrome phenotype. In this case, over-expression of TMPRSS2 might explain why Down patients are more prone to viral diseases (Influenza, SARS-CoV-1, Metapneumovirus, MERS, and SARS-CoV-2) that use ACE-2-receptor/TMPRSS2 for cell binding/cell entry. They might get more cells infected because of more protease activity and thus more easily cell entry of the viral pathogens.) was first mentioned in: De Cauwer H. The SARS-CoV-2 receptor, ACE-2, is expressed on many different cell types: implications for ACE-inhibitor- and angiotensin II receptor blocker-based cardiovascular therapies: comment. Intern Emerg Med. 2020 Nov;15(8):1581-1582. doi: 10.1007/s11739-020-02406-z. Epub 2020 Jun 20. PMID: 32564289; PMCID: PMC7305057. According to RSV (line 76) there is a recent study of a Dutch group: RSV Gold Study: this study can also be referred to in methods when you discuss the co-morbidities. : Löwensteyn YN, Phijffer EWEM, Simons JVL, Scheltema NM, Mazur NI, Nair H, Bont LJ; RSV GOLD Study Group. Respiratory Syncytial Virus-related Death in Children With Down Syndrome: The RSV GOLD Study. Pediatr Infect Dis J. 2020 Aug;39(8):665-670. doi: 10.1097/INF.0000000000002666. PMID: 32332221; PMCID: PMC7360096.

Response 1. We added the citations in the text. We also incorporated this sentence: “"However, because over-expression of TMPRSS2 explains the greater impact of viral diseases (Influenza, SARS-CoV-1, Metapneumovirus, MERS) that use ACE-2-receptor/TMPRSS2 for cell binding/cell entry on patients with DS, it is unknown to what extent the increased clinical vulnerability recently observed in DS is specific for COVID-19 [7].”

Materials and methods:
Point 2. Data extraction: line 120: 'CMBD admissions': 'hospital admissions found in CMBD' sounds better.

Response 2. Following the suggestion of the reviewer we have changed this sentence 

Point 3. line 127: here 16 years is mentioned in line 175 you write 15 years???

Response 3. Thanks for spotting this error. Patients included were older than 15 years. We excluded patients under 16 years of age as those were exceptional in the T21RS clinical survey (only 3 from these 89) obtaining 86 patients. We have now changed the text to make it clearer. “For the comparison with the non-COVID-19 pneumonias, from the 150 patients with DS we included the 86 older than 15 years infected by SARS-CoV-2 who were hospitalized due to COVID-19 (86/150;59.3%).” See lines 176-179. We also modified the text in lines 106-107 for consistency: “COVID-19 patients under 16 years of age were exceptional in the T21RS clinical survey (only 3/89). Thus, we limited the study to patients older than 15.” 

Point 4. line ??? (not mentioned) and line 287: age 40 was already mentioned in other studies: maybe also refer to in materials and methods.

Response 4. We refer to the ages in our cohort in the results section 3.1. Socio-demographic and clinical characteristics of the Spanish DS COVID-19 patients

Point 5. line 129: (see also discussion on TMPRSS2 and RSV above): why not looking for diabetes, osas, heart disease, respiratory disease. diabetes discussed in line 276 but not in methods.

Response 5. We evaluated three relevant clinical comorbidities (obesity, dementia and epilepsy/seizures), due to their association with worse outcomes for respiratory diseases, including COVID-19, in patients with DS, but also for technical reasons, as we wanted to compare those data with CMDB. In this database for selecting the cases with these diagnoses it would be necessary to identify dozens of codes including not only the principal diagnosis but secondary diagnoses in which this information is usually included and finally we decided to focus on the main diagnoses that showed statistically significance in the international study in which diabetes, heart disease, etc. did not show significant differences.

Results:
Point 6. line 167: more reporting of alzheimers disease due to line 165 older age ?

Response 6. It is possible in fact that older age explains why Alzheimer’s disease was reported significantly more in the clinicians survey, as hospitalized people were older (72% of hospitalized patients were >40). 

Point 7. line 170 a recent study in near future to be published in BMJ Open of a UK based group and already mentioned in:  

https://assets.publishing.service.gov.uk/government/uploads/system/uploads/attachment_data/file/908434/Disparities_in_the_risk_and_outcomes_of_COVID_August_2020_update.pdf

it was shown that people with mental disabilities reported less complaints like anosmia probably leading to late diagnosis of Covid. In fact,  onset of disease or reason for hospitalization was coma,seizures,....(see also line 308-309): please add to discussion.

Response 7. The reviewer points out an interesting aspect of the clinical spectrum of COVID-19 and how symptoms might be conveyed by a population with intellectual disabilities. We have made no specific mention in our paper to this point because: a) the clinical presentation of COVID-19 in patients with DS had already been reported in previous studies (particularly Huels A et al. eClinical Medicine. 2020) and b) our goal was not to discuss this clinical presentation, but to assess differences in mortality and risk factors between COVID-19 and other respiratory infections. 

Point 8. table 1: please add % in the table, it is very confusing what is SD and what is % . so also change the legend accordingly.

Response 8. Added and changed

Point 9. line 199: similar results reported in the UK based study (NHS):  as mentioned later surge capacity, ventilator allocation and end of life decisions might play a significant role. In the UK  study people with mental disabilities significantly had less ICU admissions and less ventilator access  versus controls.   

Response 9. We agree with the reviewer. In fact, this point is one of our major worries that stems from an indirect interpretation of our findings. We had already reflected on this point in lines 327-332.

Some other last questions/remarks:

Point 10.  do  you have data on how many patients with DS lived with their families or lived in institutions: sick family members  or an outbreak in the institution  might explain longer hospital stay.

Response 10.  Data on residence was summarized in SUPPL TABLE 1 (variable “living in residential care”). Unfortunately even though we have information about the percentage of institutionalized patients, our survey did not collect data on whether there have been local outbreaks in the patient’s milieu (either home or facility). 

Nevertheless, among patients admitted to hospital the number of patients living with their families were similar, 51.2% lived with their families and 48.8% in institutions, whereas the age was very different, so 72.1% were 40 years old or older and 27.9% younger than 40 years. In addition, in the group of 40 years old or older the hospital stay was shorter due to the high mortality rate associated with these patients that usually happened in the first days of the admission (figure 1).

Point 11.  it would be interesting in looking at the same data but with a group of viral pneumonias related  to TMPRSS2 versus other pathogens.

Response 11.  We thank the reviewer for this valuable suggestion, which we’ll incorporate in future work.

Round 2

Reviewer 1 Report

Ok i suggest the publication